# MULTILINGUAL ROUTING IN MIXTURE-OF-EXPERTS

**Lucas Bandarkar**[*]  **Chenyuan Yang**[§]  **Mohsen Fayyaz**  **Junlin Hu**  **Nanyun Peng**
University of California, Los Angeles     [§]Fudan University

## ABSTRACT

Mixture-of-Experts (MoE) architectures have become the key to scaling modern LLMs, yet little is understood about how their sparse routing dynamics respond to multilingual data. In this work, we analyze expert routing patterns using parallel multilingual datasets and present highly interpretable layer-wise phenomena. We find that MoE models route tokens in language-specific ways in the early and late decoder layers but exhibit significant cross-lingual routing alignment in middle layers, mirroring parameter-sharing trends observed in dense LLMs. In particular, we reveal a clear, strong correlation between a model's performance in a given language and how similarly its tokens are routed to English in these layers. Extending beyond correlation, we explore inference-time interventions that induce higher cross-lingual routing alignment. We introduce a method that steers the router by promoting middle-layer task experts frequently activated in English, and it successfully increases multilingual performance. These 1-2% gains are remarkably consistent across two evaluation tasks, three models, and 15+ languages, especially given that these simple interventions override routers of extensively trained, state-of-the-art LLMs. In comparison, interventions outside of the middle layers or targeting multilingual-specialized experts only yield performance degradation. Altogether, we present numerous findings that explain how MoEs process non-English text and demonstrate that generalization is limited by the model's ability to leverage language-universal experts in all languages.

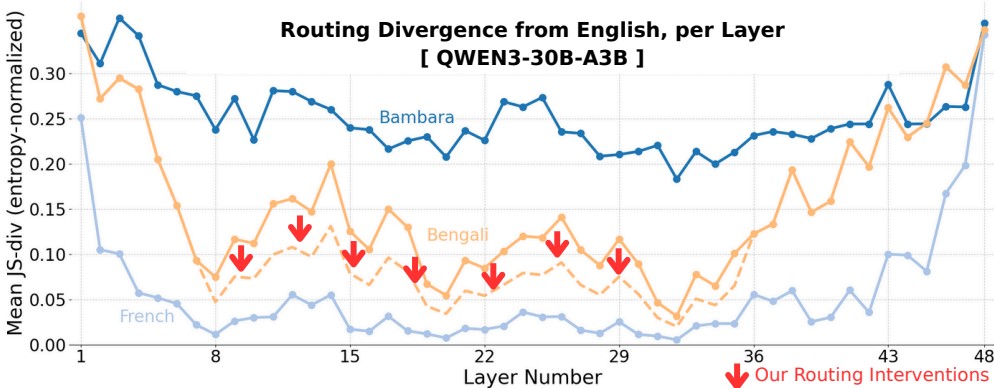

Figure 1: Visualization of the typical divergence in MoE routing weights across model layers between English and a high-, medium-, and low-resource language. There is consistently lower divergence in the middle layers, where experts are shared across languages. Languages the model does not understand (e.g. Bambara) fail to leverage similar experts as top languages. In this work, we also present a steering method that activates similar experts to English (red arrows) and results in improved multilingual generalization (e.g. an increase in MGSM-Bengali from 0.776 to 0.824).

## 1 INTRODUCTION

Sparse mixture-of-expert (MoE) architectures (Shazeer et al., 2017) are the new dominant paradigm in Large Language Models (LLMs) because they enable tremendous parameter scaling while main-

---

[*]Correspondence to `lucasbandarkar@cs.ucla.edu`

taining manageable inference costs (Artetxe et al., 2022; Du et al., 2022). In terms of interpretability, MoEs present a trade-off compared to dense models. Their sparsity enables more redundancy in parameterization (Dai et al., 2024; Li et al., 2025) and the routing mechanisms are sensitive and variable (Yang et al., 2025b). However, their discrete expert activation facilitates the analysis of which model components are responsible for the end result.

Scaling has driven remarkable progress in LLM multilingual capabilities, yet pre- and post-training are often heavily English-centric. As a result, large performance gaps remain beyond a select few languages. In recent years, significant effort has been devoted to interpreting the internal mechanisms that enable multilinguality. This has generally unveiled shared feature spaces in the middle part of the model, which are pivotal to the cross-lingual transfer of model capabilities (Kojima et al., 2024; Wendler et al., 2024; Bandarkar et al., 2025; Wu et al., 2025). However, this work has been limited to *dense* LLMs, whereas sparse activation in MoE architectures leads to different computational structures whose impact on feature representations remains unexplored.

In this work, we investigate multilingual behavior in mixture-of-experts LLMs. To begin with, a data analysis comparing routing across parallel datasets yields numerous coherent findings. Studying QWEN3-30B-A3B (Yang et al., 2025a), PHI-3.5-MOE (Abdin et al., 2024), GPT-OSS-20B OpenAI (2025), and OLMOE (Muennighoff et al., 2025), we find that, despite their sparsity, they adopt similar mechanisms as dense LLMs; leveraging language-agnostic parameters in intermediate model layers—if anything, in a clearer, more modular way. In addition, language performance is strongly correlated to its cross-lingual routing alignment to English. We further highlight how multilingual expert specialization impacts router entropy and token-to-token routing similarity.

We build upon this observation by showing that the model's ability to call upon shared experts is a key driver of multilingual performance. We investigate this via manual interventions into the MoE block's forward pass to encourage or discourage the activation of specialized experts. We explore steering the routers in different model layers, intervention strengths, and types of experts. In the end, we find that we can improve multilingual task performance when activating experts important for solving that task in English. We experiment with QWEN3, PHI-3.5-MOE, and GPT-OSS—all fully post-trained and state-of-the-art LLMs—and find that these inference-time interventions consistently yield statistically significant improvements on two tasks requiring domain knowledge, MGSM (Shi et al., 2023) and the medicine subset of GLOBAL-MMLU (Singh et al., 2025). The specific conditions under which our intervention works provides strong validation that our initial interpretability analysis uncovered verifiable MoE mechanisms for processing multilingual text.

Through this routing data analysis and the resulting intervention experiments, we demonstrate that improved expert-sharing leads to the generalization of complex capabilities. By demonstrating that simple inference-time interventions yield substantial improvements, our work reveals a vast potential for improving multilingual performance in MoE LLMs. This result motivates the development of other methods that promote cross-lingual expert sharing, such as during training.

## 2 RELATED WORK ON MULTILINGUAL LLMS

Before the massive scaling of decoder-only LLMs, smaller encoder-decoder models were subject to the *curse of multilinguality*, where adding more languages hurt performance in other languages due to limited representational capacity (Conneau et al., 2020; Pfeiffer et al., 2022). Cross-lingual embedding alignment was commonplace with such models in order to unify feature spaces and facilitate multilingual generalization (Zhou et al., 2016; Schwenk & Douze, 2017; Ouyang et al., 2021; Patra et al., 2023). But with the shift to decoder-only LLMs, this approach became no longer viable. Nonetheless, these LLMs implicitly learn shared feature representations. As noted previously, many works have concluded through different approaches that the middle decoder layers of an LLM contain joint language representations (albeit English-centric), while the first and last layers primarily map language-specific representations to and from this space (Kojima et al., 2024; Wendler et al., 2024; Tang et al., 2024; Alabi et al., 2024). As model size and training has been scaled, this phenomenon has become more evident (Chen et al., 2025) and modular (Bandarkar et al., 2025). Wu et al. (2025) finds that this semantic space extends beyond natural languages, to encompass numbers, computer languages, and different input modalities.

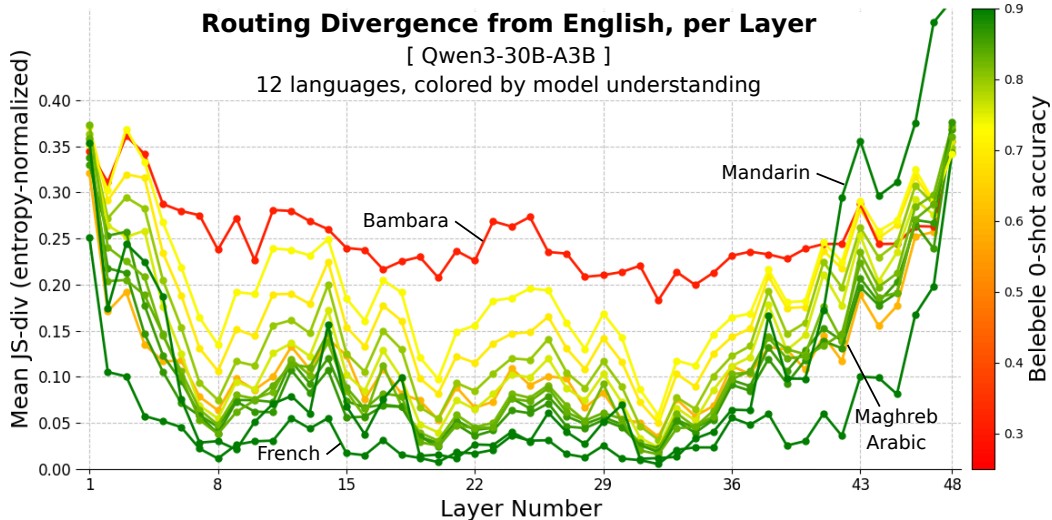

Figure 2: Visualization with more languages of routing divergence from English across model layers based on Qwen3-30B-A3B, where the U-shape can be seen for all. Each line is colored by how well the model understands that language (BELEBELE accuracy), highlighting a strong correlation between the two. We label a few notable plotted languages, but provide the same graph (along with 3 more models) colored to better distinguish languages in Appendix A.2.

Recent work finds that stronger middle-layer alignment correlates with improved multilingual performance (Kargaran et al., 2025; Ravisankar et al., 2025). This relationship appears causal: multilingual performance improves when they are explicitly steered toward language-shared representations (Mahmoud et al., 2025; Lu et al., 2025) or away from language-specific representations (Lim et al., 2025; Zhao et al., 2025). Prompting methods that encourage the use of English as a pivot language can also boost cross-lingual transfer (Shi et al., 2023; Zhang et al., 2024; Yong et al., 2025). By design, LCM team et al. (2024) introduces an LLM that autoregresses over language-neutral "concept" embeddings instead of subword tokens, which exhibits strong multilingual generalization.

Bridging multilingual research and MoEs, recent works leverage MoE modularity for massively multilingual machine translation (NLLB Team et al., 2022; Zhao et al., 2024). In LLMs, Zheng et al. (2025) scales multilinguality through MoE upcycling (Komatsuzaki et al., 2023) in final layers.

## 3  MIXTURE-OF-EXPERTS PRELIMINARIES

MoE LLMs differ from traditional decoder-only transformer architectures by replacing the multi-layer perceptron (MLP) component of each model layer with $E$ MLPs, referred to as "experts". For each input token, a router (or "gating network") calculates a set of logits and sends the token embedding to the top-$K$ experts only. The $K$ output hidden states are then aggregated, typically via a weighted sum. MoE models are often trained with an auxiliary load-balancing loss (Shazeer et al., 2017; Fedus et al., 2022) that penalizes uneven expert utilization, introducing some redundancy in expert specialization. Token-to-expert mappings are established early in pretraining and are often independent of the surrounding context (Xue et al., 2024).

## 4  INTERPRETABILITY ANALYSIS

### 4.1  DATA

For this analysis, we primarily use the FLORES-200 translation dataset (Goyal et al., 2022) because of its parallel texts and inclusion of many diverse languages. While no dataset can truly be without domain or style, we use FLORES and its wide array of topics to represent the baseline, *generic* domain. Conveniently, FLORES has an associated reading comprehension evaluation dataset, BELE-BELE (Bandarkar et al., 2024), which we use to tie in language performance. We carefully select a

subset of 12 languages (plus English), diverse in scripts, families, and resource-levels that allow us to explore numerous relationships (See the list in the Appendix A.3).

## 4.2 MODELS

We look at four prominent open-source MoE LLMs: OLMOE (Muennighoff et al., 2025), QWEN3-30B-A3B (Yang et al., 2025a), PHI-3.5-MOE (Abdin et al., 2024), GPT-OSS-20B (OpenAI, 2025). All have been trained primarily on English data, but the technical reports of QWEN3, GPT-OSS, and PHI-3.5-MOE emphasize their multilingual capabilities. Presumably, these three have been pre- and post-trained on significant non-English data. Meanwhile, the older and smaller OLMOE is English-only, exhibiting much poorer multilingual performance. These models all differ in their architectural width, sparsity, and depth. We provide model details and checkpoint specifics in Appendix A.1.

## 4.3 ROUTING DIVERGENCE

We begin by collecting routing data on FLORES for each language across layers. Due to the difficulty of cross-lingual token alignment, we average the post-softmax routing weights across tokens to obtain each sequence's *expert importance distribution*. Given a language *lang* and model layer $l$:

- let $E$ be the number of experts in the Mixture-of-Experts (MoE) layer.
- let $N$ be the number of sequences in the corpus.
- let $L_i$ be the sequence length (number of tokens) of the $i^{\text{th}}$ sequence.
- let $p_{i,t}^{(\text{lang},l)}$ be the routing weights for the $t^{\text{th}}$ token of the $i^{\text{th}}$ sequence from language *lang*, at layer $l$. This is an $E$-dimensional probability; If $z$ are the logits, then $p_{i,t}^{(\text{lang},l)} = \text{softmax}(z_{i,t}^{(\text{lang},l)})$.

$$q_i^{(\text{lang},l)} = \frac{1}{L_i} \sum_{t=1}^{L_i} p_{i,t}^{(\text{lang},l)} \qquad \in [0,1]^E \tag{1}$$

The expert importance $q$ for the $i^{\text{th}}$ sample is the mean-pooled routing weights across tokens ($q \in [0,1]^E$ and $\sum q = 1$). We consider alternatives, such as averaging discrete activation counts rather than routing probabilities, but these yield sharper, higher-variance distributions that are more difficult to mean-pool. Another option is to use only the last token's routing weight (or average the last few), as is sometimes done with hidden states. However, routing weights vary more strongly across tokens than hidden states, due to factors like part-of-speech, token type, and positional context. Therefore, sequence-wide weight-averaging provides a more stable and representative measure of routing behavior.

For each non-English sequence, we use *entropy-normalized* Jensen-Shannon divergence ($D_{\text{H-JS}}$) to compare its expert importance distribution to that of its paired English sequence. Routing entropy consistently decreases across model layers and therefore needs to be accounted for when comparing JS-divergence (a symmetric variant of KL-divergence) across layers. We revisit this trend in Section 4.4 and detail our entropy normalization in Appendix A.4. Finally, we average these divergences across all sequences in the corpus to get a metric for routing divergence from English for each layer and language.

$$\text{Div}^{(\text{lang},l)} = \frac{1}{N} \sum_{i=1}^{N} D_{\text{H-JS}}(q_i^{(\text{eng},l)} || q_i^{(\text{lang},l)}) \qquad \in [0,1] \tag{2}$$

This designed metric reveals highly interpretable patterns across languages, models, and layers.

> **Finding 1**
>
> For all languages, there is much higher routing divergence from English in the first and last layers than in the intermediate layers. The overall trend is this U-shape for all languages (See Figure 2).

And while this trend is the most pronounced and least noisy for QWEN3, this general trend is common to all four models evaluated (See Appendix A.2). As mentioned, OLMOE has very poor multilingual capabilities and this could explain why the big majority of languages studied do not exhibit this U-shape (See Figure A.2). However, French (fra) and Chinese (zho), high-resource languages it can somewhat process, display this U-shape clearly. GPT-OSS (Figure A.3) displays this trend clearly and is the only one where divergence is higher in the first layers than the last. PHI-3.5-MOE (Figure A.4) displays this trend, but only if you exclude the first two layers—PHI-3.5-MOE

perplexingly activates the same few experts in the first layer for all languages. We are unable to find an explanation for this, but this could imply very poor load-balancing.

This aligns with findings from dense LLMs that reveal language-universal representation spaces in the middle layers of LLMs. If the representations are aligned, the routing would also be. This means that other factors, perhaps semantic ones rather than lexical ones, determine routing here. Generally, these MoE LLMs have implicitly learned to call upon similar experts across languages. While English is the main pivot language for all models studied, we find similar trends when graphing with another focus language. Trends are more flattened if lower-resource languages are used as the focus.

> **Finding 2**
>
> We find a strong correlation between cross-lingual routing alignment in intermediate layers and language performance. See Figure 2 for a visualization of this phenomenon on QWEN3.

This figure displays a strikingly strong relationship between a model's ability in a language (line color) and how aligned its routing is with English. This is true across the model layers except the first and last ones. Generally, the highest-resource languages form the strongest U-shape, with very low routing divergence in the middle layers. For Bambara, an example for a language we know the models are all very poor at (near-random BELEBELE performance), the LLMs fail to map its inputs to this semantic space, maintaining high routing divergence throughout. In the middle layers of all models, the correlation coefficient $r$ between the routing divergence from English and BELEBELE accuracy is always strong. For OLMoE, $r \in [-0.95, -0.80]$ for *all* middle layers. Meanwhile, GPT-OSS is the weakest ($r \in [-0.40, -0.60]$), with PHI-3.5-MoE and QWEN3 in between. Recall that BELEBELE directly evaluates understanding on those same FLORES passages.

Across all layers, language similarity is also correlated with routing similarity. This is very expected in the first and last layers, where token overlap and structural similarity lends itself to shared parameterization. Even so, the impact of language families continues into the middle layers where we find that even here, related language pairs like (Bengali, Assamese) or (Romanized Arabic, MSA) have much lower routing divergence between them than unrelated pairs like (Bengali, French) and (Oriya, Serbian). And while this confounds our analysis relating language ability and routing alignment, we find that language relationships only explain a small part of the trends. Visualizing divergence from Chinese, French, or other high-resource languages (instead of English) yields very similar plots, although the slight attenuation of the trends reflect the English-centricity of the shared representations.

We additionally explore domains instead of languages to compare. We select AlpaCare-MedInstruct (Zhang et al., 2023) to represent the medical domain, GSM8K-Instruct (Cobbe et al., 2021) the mathematical domain, and the English FLORES split as the baseline . For these domains, routing divergence from the generic domain exhibits the *opposite* pattern: higher divergence exists in the middle layers (more of a ∩-shape). However, these patterns are weaker, as domains are less different than languages. Nevertheless, this suggests separation of parameterization between multilinguality and task-specific capabilities, as has been observed in dense

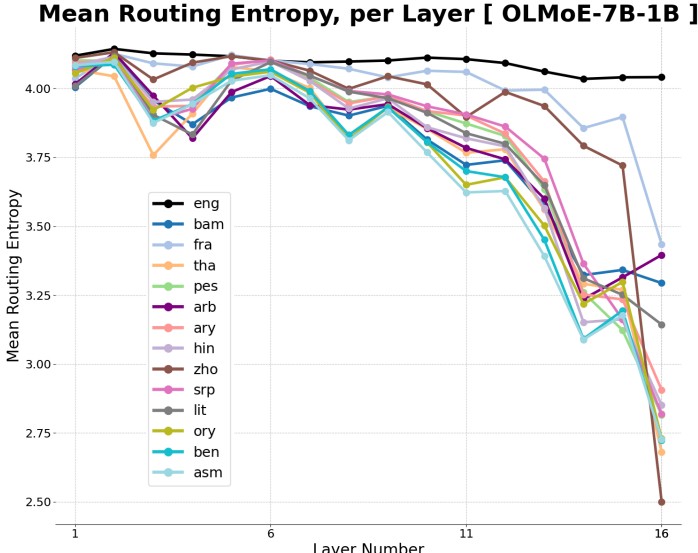

Figure 3: Routing Entropy per Layer for OLMoE.

models (Bandarkar et al., 2025). We revisit this language-task *modularity* (Choenni et al., 2024; Bandarkar & Peng, 2025) in Section 5.2, as it is fundamental to our intervention methodology.

## 4.4 ROUTING ENTROPY AND CONSISTENCY

> **Finding 3**
>
> Routing entropy decreases (in other words, routing confidence increases) across model layers for all languages, but this decrease happens at a much higher rate for non-English languages.

We calculate the entropy $H$ of each routing weight distribution and then average across tokens in the datasets. As tokens pass through the model, routers increasingly know which experts to send them to, perhaps because representations become more refined or experts more diverse (Lo et al., 2025). This lowering entropy occurs for English, but is much more pronounced for non-English languages, as can be seen for OLMoE in Figure 3. For non-English languages in particular, the final layer displays a major drop. While the entropy graphs look quite different for each model (See Appendix A.5), these broader trends are visible for all. A likely explanation is a small set of experts specialized for non-English generation in the final layers. Meanwhile, English has a broader pool of possible experts to decide from, resulting in higher routing entropy.

We additionally analyze the token-to-token routing variance across languages and layers. To do so, we randomly sample 500 pairs of tokens per sequence and take the Jaccard similarity of the two sets of activated experts. This gives a robust estimate for the expected similarity across all $2^L$ token pairs. We then average across sequences, giving a measure of *intra*-sequence routing agreement for each layer. We show this metric for PHI-3.5-MOE with a subset of languages in Figure 4 as an example. Other models and languages display very similar patterns.

> **Finding 4**
>
> For non-English languages, there is generally higher routing consistency across tokens *within a sequence* than in English. In the last layers, it is **much** higher than English.

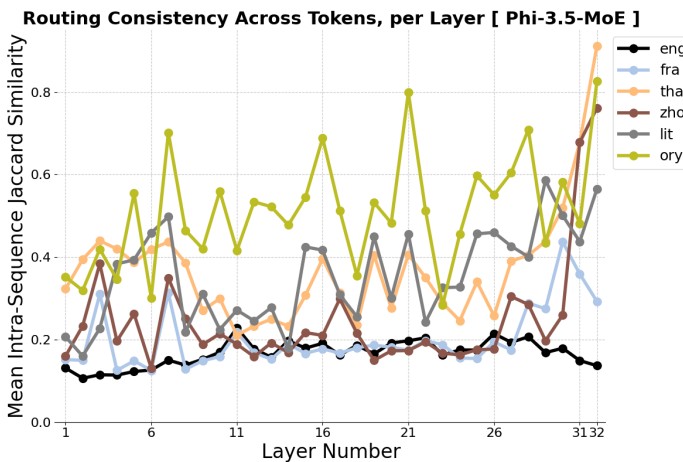

Figure 4: Token routing consistency (within a sequence), across layers in PHI-3.5-MOE.

Routing consistency, itself, also correlates with performance, with the lowest resource languages having the highest token-to-token agreement. High-resource languages have lower consistency, but still mostly above that of English across model layers. This can be explained by the English-heavy pretraining using a load-balancing loss, which would lead to a large number of specialized experts for English tokens. In contrast, multilingual tokens rely on fewer experts, resulting in reduced token-to-token variation. In the last layers, the models will consistently send tokens in the same non-English language to the same expert. These consistency metrics further support the existence of many fewer multilingual experts in the later layers than those available for English.

## 5 INTERVENTION METHODOLOGY

### 5.1 EVALUATION TASKS AND DATA

We choose as our target evaluation tasks the multilingual mathematical reasoning benchmark, MGSM (Shi et al., 2023), and the medicine subset of the multiple-choice benchmark, GLOBAL-MMLU (Singh et al., 2025). Both are fully parallel test sets that require domain-specific knowledge and reasoning, and therefore present a good evaluation of cross-lingual ability transfer in comparison to FLORES and BELEBELE, which serve more as pure linguistic signals . To identify math domain experts, we use GSM8K-Instruct (Chen et al., 2024), which constitutes the training set of GSM8K

(Cobbe et al., 2021) augmented with instructions. Once again, we use the AlpaCare MedInstruct dataset (Zhang et al., 2023) to represent the medical domain. We note that GSM8K is distributionally identical to (English) MGSM, while MedInstruct is only similar to the evaluation data in broad domain. We continue to use FLORES as the baseline (see Section 4.1).

## 5.2 EXPERT IDENTIFICATION

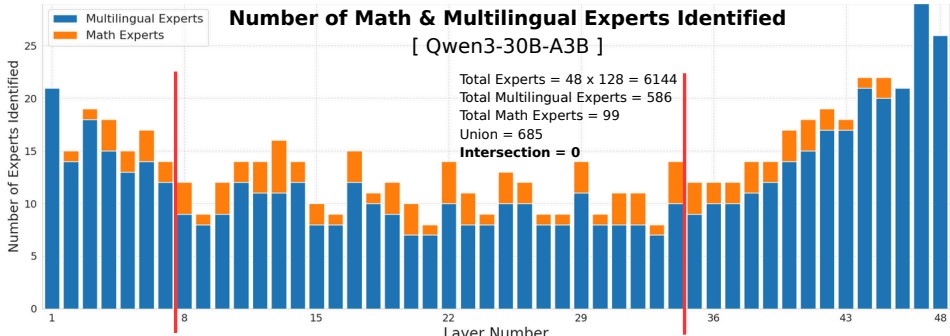

Figure 5: Plot of the Number of Identified Experts per Layer, with $\tau = 0.3$ for QWEN3. The red vertical bars delimit the region in which we intervene. A different view is displayed in Appendix A.8

Although the goal of Section 4 was to understand layer-wise patterns, the goal here is to identify specialization in individual experts. To do so, we use discrete activation counts instead of routing weights to better discern the most responsible experts. Similarly to Fayyaz et al. (2026), we calculate the *relative* frequency of activation, that is the proportion of tokens that an expert figured in the top-k, $\boldsymbol{a}_i/L_i \in [0,1]^E$. While that work uses paired samples, we cannot pair our data across domains and thus average this activation proportion across sequences in the corpus. Then, for each expert, we take the difference $\boldsymbol{\Delta}$ in those averages between the task or language dataset and the baseline. Concretely:

$$\boldsymbol{\Delta} = \frac{1}{N^{(1)}} \sum_{i=1}^{N^{(1)}} \frac{\boldsymbol{a}_i}{L_i} - \frac{1}{N^{(2)}} \sum_{j=1}^{N^{(2)}} \frac{\boldsymbol{a}_j}{L_j} \quad \in [-1,1]^E \tag{3}$$

We find that this method helps clearly identify specialized experts. This is because the resulting $\boldsymbol{\Delta}$s are heavily *right-skewed*. The large majority of experts are around zero, or right below, while a small percentage of experts have strong positive values (See Appendix A.7 for a visualization). These experts, activated much more often on the domain or language dataset compared to the baseline, are therefore specialized. This also shows that, empirically, FLORES is sufficiently "generic" to serve as a baseline. To select the experts to intervene on, we use a tunable positive-valued threshold $\tau$. Therefore, the $k^{\text{th}}$ expert is selected for intervention if $\boldsymbol{\Delta}_k > \tau$. Given the high number of languages available, we define a multilingual expert as one where *for any* language [1] , $\boldsymbol{\Delta}_k > \tau$.

> **Finding 5**
> There is **no** overlap between multilingual-specialized experts and task (math or medicine) experts.

We find multilingual experts in all layers, but as expected, we find a lower prevalence in the middle (See Figure 5 for an example). The *degree* of specialization is also much lower here (lower $\Delta_k$-values). Math and medicine experts are more evenly distributed across the model and generally have lower $\Delta_k$-values. If $\tau \geq 0.3$, we detect absolutely no expert simultaneously specialized for a task and multilinguality across all four models. In other words, the set of experts that are activated more in non-English languages is completely separate from those activated more in the math and medicine domains than the general domain. This is consistent with the routing divergence trends and is a very convincing display of language-task modularity in MoE LLMs.

This finding provides striking empirical support for the "functional dissociation" of language and thought in LLMs, as proposed by Mahowald et al. (2024). The clear segregation between multilingual experts and task-specific experts suggests that the model's internal architecture reflects a modular distinction between formal linguistic competence (i.e. in diverse languages) and functional competence (i.e. for a task).

---

[1] We do investigate with identifying individual-language experts, but intervention results end up the same.

## 5.3 ROUTING INTERVENTIONS

The above produces a set of experts to boost ($\mathbb{A}^+$) or suppress ($\mathbb{A}^-$). Then, during the forward pass through the MoE block, our method intercepts the router logits and alters them. This not only has an impact on the sparse activation of experts, but also how heavily each output is weighed during aggregation. Thus, we intervene prior to the softmax operation to not destabilize this weighted sum.

**Soft intervention** Our first style of intervention simply steers the original logit of the target expert. Because every router produces logits in a very different range, we elect to steer by adding/subtracting values ($\lambda$) proportional to the standard deviation of all $E$ logits, $s(\boldsymbol{z})$. Our approach is quite different from Wang et al. (2025), which steers by rather multiplying the weights by a factor. Empirically, we find smaller $|\lambda| \leq 1.0$ to work best. Concretely, to steer the $k^{\text{th}}$ expert (in $\mathbb{A}^+$ or $\mathbb{A}^-$):

$$\boldsymbol{z}'_k \leftarrow \boldsymbol{z}_k + \lambda \cdot s(\boldsymbol{z}) \tag{4}$$

**Hard intervention** We replicate the *force*-activation and -deactivation method from Fayyaz et al. (2026), which sets a logit to the maximum or minimum value among all experts on a given token, forcing its selection or non-selection. We also add a random perturbation $\varepsilon$ for edge cases (See Appendix A.6). So, if expert $k \in \mathbb{A}^+$:

$$\boldsymbol{z}'_k \leftarrow \max(\boldsymbol{z}) + \varepsilon, \qquad \varepsilon \sim \mathcal{N}(0, 10^{-6}) \tag{5}$$

Or rather $\min(\boldsymbol{z})$ if $k \in \mathbb{A}^-$. We note that deactivation is a less significant intervention than activation: deactivation removes one of many options, rather than forcing one of the few activations. Either way, the intervened routing is still far from homogenized across languages. Even hard intervention only forces 1 or 2 experts into the top-K selection (where K is typically 4 or 8). The router still produces a full logit distribution for every token and selects most experts as before. The intervention is simply a targeted nudge to ensure the task-relevant experts are strongly considered.

## 5.4 INTERVENTION EXPLORATION

The search space of possible such interventions is large, given (1) the choice of model layers, (2) the types of specialized experts to target (task or multilingual experts), (3) the threshold $\tau$ that controls expert selection, and finally (4) the direction and (5) strength of intervention. To limit the exponentially large search space of model layers (1), we leverage our layer-wise interpretability analysis of where strong relationships exist between alignment and (linguistic) performance, i.e., the middle layers. Specifically, we use the divergence graphs (Figure 2, Appendix A.2) to demarcate which layers constitute this middle zone (See "Target layers" in Table 1). For example, we hypothesize that boosting alignment would help most in QWEN3's layers 8 to 35 (one-indexed).

We started by exploring the milder deactivation of experts rather than force-activation. As baselines, we deactivated multilingual or task experts in all layers or in random subsets, which led to substantial performance degradation. Leveraging our hypothesis, we found that a significant number of multilingual experts in the middle layers could be suppressed without causing major performance degradation. Similarly, task experts in the early and late layers could be deactivated with minimal negative impact. Deactivating in the exact opposite layers, for each, led to a large drop. While deactivation only led to worse performance, these patterns resonated with our hypothesis of implicit language-task modularity. This informed our subsequent strategy.

Generally speaking, random or even slightly suboptimal interventions to the router lead to performance degradation. This highlights the sensitivity of intervening in heavily-trained MoE routers and the potential to over-/under-weigh particular experts during inference.

Table 1: Summary of Intervention Results. Target layers are the model layers where the intervention takes place. The expert-selection threshold $\tau$ and intervention method are described in Section 5. Given the target layers and $\tau$-value, we provide the number of experts selected for steering.

| Model name | Total layers | Target layers | $\tau$ ($\Delta_k$ thresh.) | Interven. method | #Experts selected | Non-Eng AVG original | Non-Eng AVG intervened |
|---|---|---|---|---|---|---|---|
| *MGSM* | | | *10 non-English languages, 250 samples, 2-shot exact-match* $\uparrow$ | | | | |
| QWEN3-30B-A3B | 48 | (8,35) | 0.4 | soft,$\lambda$=0.5 | 22 | 76.4% | **78.0%** |
| PHI-3.5-MOE | 32 | (8,17) | 0.3 | soft,$\lambda$=0.5 | 12 | 57.5% | **58.9%** |
| GPT-OSS-20B | 24 | (4,19) | 0.3 | hard | 9 | 68.9% | **71.5%** |
| *Global-MMLU, Medical Subset* | | | *13 non-English languages, 420 samples, 0-shot accuracy* $\uparrow$ | | | | |
| QWEN3-30B-A3B | 48 | (8,35) | 0.5 | hard | 23 | 68.2% | **69.1%** |
| PHI-3.5-MOE | 32 | (8,17) | 0.25 | soft,$\lambda$=0.5 | 2 | 57.8% | **58.8%** |
| GPT-OSS-20B | 24 | (4,19) | 0.3 | soft,$\lambda$=0.5 | 6 | 63.8% | **64.5%** |

## 6 INTERVENTION RESULTS

Given the lack of improvement from any deactivation schema, we ultimately turn to the more invasive strategy of boosting or force-activating specialized experts, which leads to compelling findings:

> **Finding 6**
> For all models and tasks, steering the router to use the same middle-layer experts that it activates for a task in English leads to a statistically significant improvement in multilingual performance.

As discussed, we identify task experts using English in-domain data and the above methodology. Then we intervene to encourage or force the activation of such experts in the middle layers when evaluating on multilingual splits of the benchmark. As displayed in Tables 1 and 2, this intervention method is very consistent in its positive increase for 3 models and 2 evaluation tasks. These gains can be seen across the diverse range of languages, even a bit more pronounced for lower-resource languages. While the magnitude is modest (1-2 accuracy points), this is substantial given the simplicity of the test-time intervention. It is also statistically significant when considered across languages. The gains for medicine are less pronounced than for math, likely because the dataset for identification is less directly aligned to the evaluation data.

**Hyperparameters** The models all differed in their MoE configurations (See Appendix 4.2) and magnitudes of $\Delta$-values, requiring model-specific tuning for the expert-selection threshold $\tau$ and the strength of intervention (e.g., hard or soft). For soft interventions, $|\lambda| = 0.5$ tended to be the most successful, as larger $\lambda$ likely disrupted the weights for aggregation significantly. Generally, intervening on a very small number of targeted experts leads to improvements. QWEN3 requires the strictest selection threshold $\tau$, while PHI-3.5-MOE works best with minimal interventions (though with 2 experts-per-token, each intervention has much greater impact). For GPT-OSS on MGSM, the effectiveness of *hard*-activation with $\tau = 0.3$ is surprising because here, an expert active on as low as 35% of domain tokens in English would be force-activated on *all* tokens during evaluation.

**Sensitivity to Target Layers** Despite the above differences across models, the sensitivity to target layers far exceeds that to these tunable hyperparameters for all models. We leveraged our divergence graphs (Figure 2, Appendix A.2) to determine the layers, and using layers slightly outside of this zone led to significant performance degradation rather than improvement. This strict delimitation is notable; it suggests that while middle layers are naturally language-agnostic, the routing differences elsewhere represent meaningful specialization that our intervention merely disrupts. In addition, these results also validate the efficacy of the visualizations from the routing analysis in revealing these boundaries. Combining with other interventions, such as activating multilingual experts in the top and bottom layers, tends to nullify the gains. In the end, our experiments show that only by targeting a small number of task-specific experts in these precise, language-universal middle layers can we achieve consistent positive gains in multilingual performance.

Overall, the baseline failures and number of conditions under which these interventions do not work highlight the difficulty of intervening in heavily-trained MoE routers without hurting performance.

Table 2: Per-Language Intervention Results. Intervention specifics are provided in Table 1.

**MGSM**                                  250 samples, 2-shot exact-match ↑

| Language | GPT-OSS-20B base | intervened | | QWEN3-30B-A3B base | intervened | | PHI-3.5-MOE base | intervened | |
|---|---|---|---|---|---|---|---|---|---|
| en | 89.6% | 89.2% | (-0.4) | 95.2% | 94.8% | (-0.4) | 88.0% | 85.6% | (-2.4) |
| bn* | 56.0% | 57.6% | (+1.6) | 77.6% | 79.6% | (+2.0) | 20.8% | 23.2% | (+2.4) |
| de | 69.2% | 76.8% | (+7.6) | 82.4% | 83.2% | (+0.8) | 79.2% | 81.6% | (+2.4) |
| es | 76.4% | 78.8% | (+2.4) | 89.2% | 89.2% | (0.0) | 85.6% | 85.6% | (0.0) |
| fr | 71.6% | 71.6% | (0.0) | 78.8% | 82.4% | (+3.6) | 70.0% | 69.2% | (-0.8) |
| ja | 77.6% | 76.8% | (-0.8) | 80.0% | 82.0% | (+2.0) | 75.6% | 77.2% | (+1.6) |
| ru | 70.4% | 76.0% | (+5.6) | 78.0% | 82.0% | (+4.0) | 79.2% | 78.8% | (-0.4) |
| sw* | 52.4% | 62.0% | (+9.6) | 48.4% | 51.6% | (+3.2) | 19.6% | 20.8% | (+1.2) |
| te* | 71.6% | 74.0% | (+2.4) | 62.0% | 62.4% | (+0.4) | 4.0% | 4.4% | (+0.4) |
| th | 58.0% | 58.4% | (+0.4) | 84.8% | 80.4% | (-4.4) | 65.2% | 68.4% | (+3.2) |
| zh | 86.0% | 83.2% | (-2.8) | 83.2% | 86.8% | (+3.6) | 76.0% | 79.6% | (+3.6) |
| non-en AVG | 68.9% | 71.5% | (+2.6) | 76.4% | 78.0% | (+1.5) | 57.5% | 58.9% | (+1.4) |
| low-res * AVG | 60.0% | 64.5% | (+4.5) | 62.7% | 64.5% | (+1.9) | 14.8% | 16.1% | (+1.3) |

**GLOBAL-MMLU, Medicine Subset**                      420 samples, 0-shot accuracy ↑

| Language | base | intervened | | base | intervened | | base | intervened | |
|---|---|---|---|---|---|---|---|---|---|
| en | 79.0% | 78.1% | (-0.9) | 82.4% | 83.1% | (+0.7) | 75.5% | 75.5% | (0.0) |
| ar | 58.1% | 58.8% | (+0.7) | 64.5% | 64.9% | (+0.4) | 55.2% | 56.9% | (+1.7) |
| bn* | 63.3% | 64.7% | (+1.4) | 63.6% | 65.5% | (+1.9) | 39.0% | 40.5% | (+1.5) |
| de | 71.0% | 70.4% | (-0.6) | 75.7% | 75.2% | (-0.5) | 68.6% | 70.2% | (+1.6) |
| es | 71.2% | 72.1% | (+0.9) | 76.4% | 78.6% | (+2.2) | 70.5% | 71.9% | (+1.4) |
| fr | 71.4% | 71.9% | (+0.5) | 77.9% | 79.3% | (+1.4) | 71.7% | 73.1% | (+1.4) |
| hi | 64.0% | 63.6% | (-0.4) | 66.2% | 66.3% | (+0.1) | 55.2% | 53.1% | (-2.1) |
| id | 66.7% | 67.4% | (+0.7) | 75.2% | 75.7% | (+0.5) | 66.0% | 68.3% | (+2.3) |
| it | 67.1% | 67.1% | (+0.0) | 76.7% | 76.6% | (-0.1) | 71.4% | 71.4% | (+0.0) |
| ja | 65.5% | 67.1% | (+1.6) | 72.9% | 72.6% | (-0.3) | 63.8% | 64.5% | (+0.7) |
| ko | 60.7% | 61.9% | (+1.2) | 68.3% | 68.8% | (+0.5) | 56.7% | 55.7% | (-1.0) |
| pt | 69.3% | 69.8% | (+0.5) | 75.7% | 77.6% | (+1.9) | 50.7% | 51.7% | (+1.0) |
| sw* | 50.2% | 51.8% | (+1.6) | 43.6% | 46.4% | (+2.8) | 40.5% | 41.9% | (+1.4) |
| yo* | 46.2% | 47.6% | (+1.4) | 42.1% | 42.6% | (+0.5) | 40.0% | 42.9% | (+2.9) |
| zh | 68.3% | 68.3% | (0.0) | 76.0% | 77.1% | (+1.1) | 59.8% | 60.5% | (+0.7) |
| non-en AVG | 63.8% | 64.5% | (+0.7) | 68.2% | 69.1% | (+0.9) | 57.8% | 58.8% | (+1.0) |
| low-res * AVG | 53.2% | 54.7% | (+1.5) | 49.8% | 51.5% | (+1.7) | 39.8% | 41.8% | (+1.9) |

As a result, the consistent improvements from our small-scale test-time intervention are notable. The layer-wise conditions under which it happens imply that the inability of such MoE LLMs to activate similar middle-layer task-experts across languages is a limitation for cross-lingual transfer.

## 7 CONCLUSION AND FUTURE WORK

All our analyses of sparse activation patterns converge on a key finding: the most important multilingual specialization of MoE experts occurs in early and late model layers, with experts in the middle layers serving as language-universal mechanisms for multilingual generalization. The strongest evidence for multilingual specialization exists in the last layers, where we find lower entropy and high token-to-token consistency in non-English languages. When identifying experts specialized for two other domains, math and medicine, we surprisingly find full separation between the sets of experts for those and for multilinguality. Our manual interventions that steer routers to replicate English activation patterns yield consistent multilingual improvements. The specific conditions under which these interventions lead to positive improvement validate our initial analysis and highlight the existence of the model's learned mechanism to attempt to leverage similar experts in the middle layers, even if not always successful. These results suggest a causal relationship between cross-lingual routing alignment and cross-lingual transfer. These findings collectively motivate future work on methods that enhance cross-lingual routing alignment and the sharing of specialized experts, such as during training. Additionally, the distinct and modular separability of parameters between language-shared and language-specific functions suggests opportunities for architectural or training approaches that exploit this natural division.

REPRODUCIBILITY

All details for reproducibility have been provided in Sections 4, 5, and 6, including but not limited to: model checkpoints, evaluation details, formulas, metrics, expert identification method, and intervention methods. Certain details have been presented in the Appendix (A.1, A.4, and A.6), but are referenced from the main text. The implemention of the intervention method was done with vLLM (Kwon et al., 2023) and evaluations with the LM Evaluation Harness (Gao et al., 2024).

ACKNOWLEDGMENTS

The authors acknowledge Tanmay Parekh, Sheriff Issaka, and Yunzhi Yao for valuable discussions at UCLA during this work.

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

# A APPENDIX

## A.1 MODEL DETAILS

Table A.1: Details of the MoE LLMs Discussed

| Model | Total Params | Active Params | Model Layers | Num. Experts | Active Experts | Checkpoint & Citation |
|---|---|---|---|---|---|---|
| OLMOE | 7B | 1.0B | 16 | 64 | 8 | OLMoE-1B-7B-0125-Instruct Muennighoff et al. (2025) |
| PHI-3.5-MOE | 42B | 3.8B | 32 | 16 | 2 | Phi-3.5-MoE-instruct Abdin et al. (2024) |
| GPT-OSS | 20B | 3.6B | 24 | 32 | 4 | gpt-oss-20B OpenAI (2025) |
| QWEN3 | 31B | 3.3B | 48 | 128 | 8 | Qwen3-30B-A3B Yang et al. (2025a) |

## A.2 ROUTING DIVERGENCE PLOTS FOR ALL MODELS

See Appendix A.3 for language code mappings.

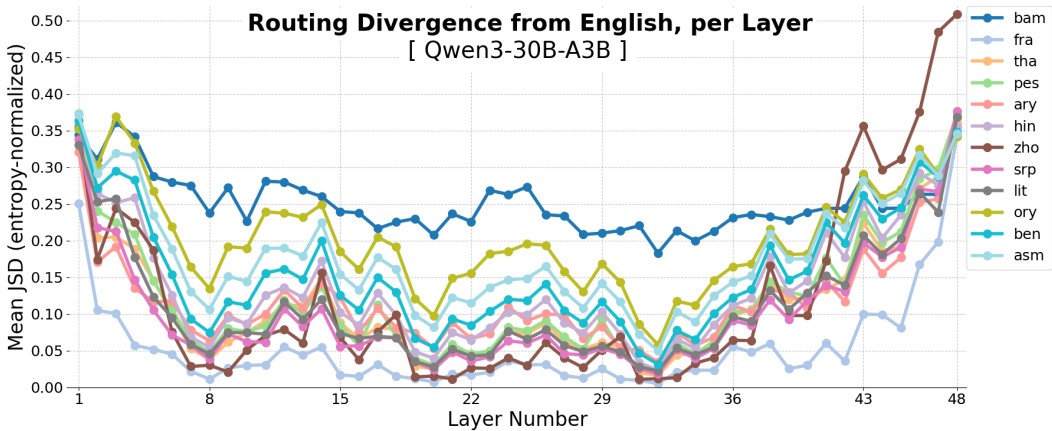

Figure A.1: Mean entropy-normalized JS-Div per OLMOE layer for 12 non-English languages. This is the same plot as Figure 2, simply colored for language labeling.

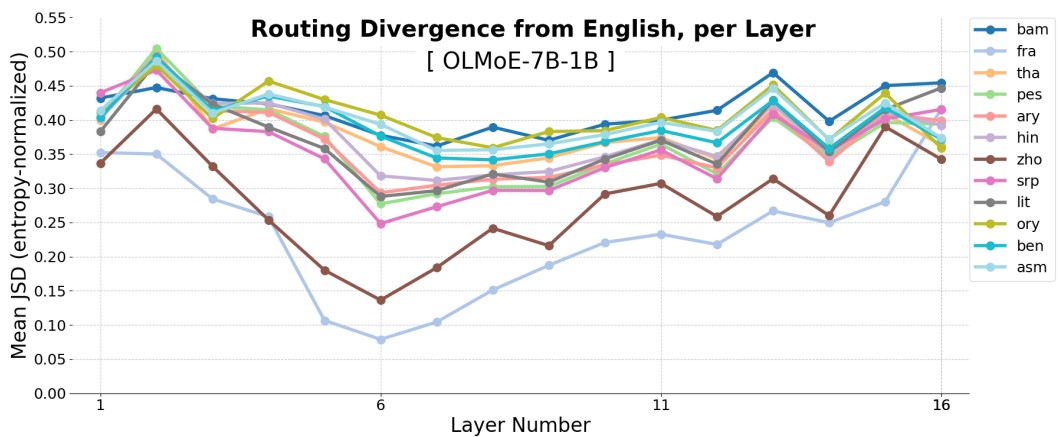

Figure A.2: Mean entropy-normalized JS-Div per OLMOE layer for 12 non-English languages. We note OLMOE's poor multilingual capabilities. For languages OLMOE does handle reasonably, French and Chinese, the U-shape is still visible.

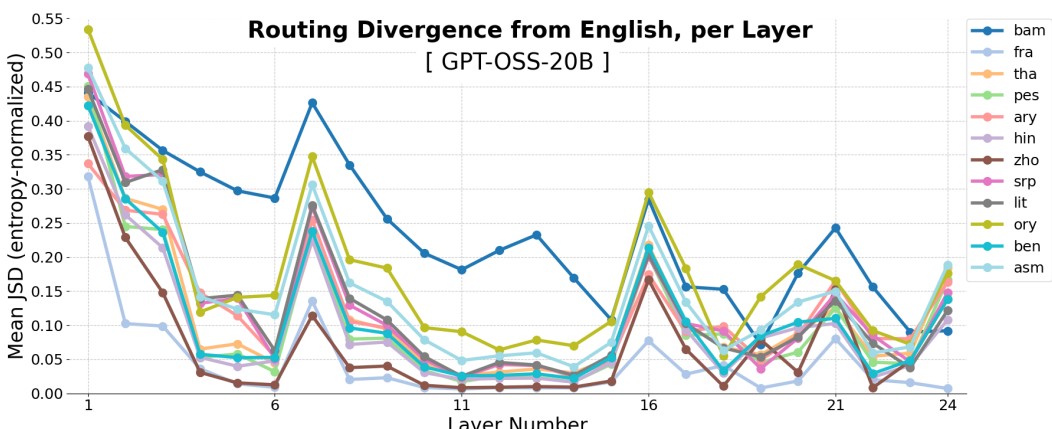

Figure A.3: Mean entropy-normalized JS-Div per GPT-OSSlayer for 12 non-English languages.

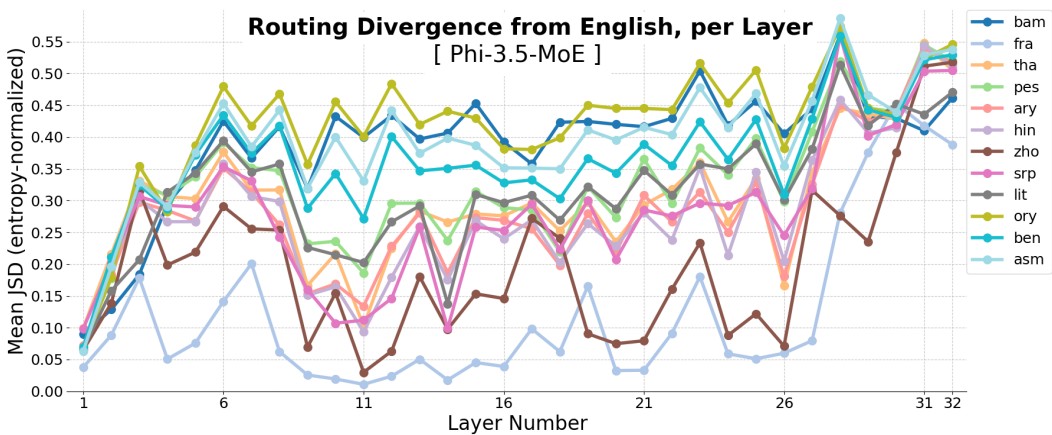

Figure A.4: Mean entropy-normalized JS-Div per PHI-3.5-MOE layer for 12 non-English languages. Compared to the others, PHI-3.5-MOE does not display the same U-shape, as the first few layers surprisingly have very low divergence, especially the first layer. We verified and saw that a small subset of available experts were being called for all languages in layer 1, which is behavior that requires further investigation.

## A.3 LANGUAGE CODE INDEX

Table A.2: The FLORES codes are used by FLORES and BELEBELE, while the 2-letter codes are used by MGSM and GLOBAL-MMLU. Our plots display the 3-letter code for brevity. We omit some abbreviations in order clarify which languages were used in which component of this work.

| FLORES Code | Language | Script | 2-letter (evaluations) | 3-letter (analysis) |
|---|---|---|---|---|
| arb_Arab | Modern Standard Arabic | Arab | ar | arb |
| asm_Beng | Assamese | Beng | | asm |
| bam_Latn | Bambara | Latn | | bam |
| ben_Beng | Bengali | Beng | bn | ben |
| deu_Latn | German | Latn | de | |
| eng_Latn | English | Latn | en | |
| spa_Latn | Spanish | Latn | es | |
| fra_Latn | French | Latn | fr | fra |
| hin_Deva | Hindi | Deva | hi | hin |
| ind_Latn | Indonesian | Latn | id | |
| ita_Latn | Italian | Latn | it | |
| jpn_Jpan | Japanese | Jpan | ja | |
| kor_Hang | Korean | Hang | ko | |
| lit_Latn | Lithuanian | Latn | | lit |
| ory_Orya | Odia | Orya | | ory |
| pes_Arab | Western Persian | Arab | | pes |
| por_Latn | Portuguese | Latn | pt | |
| rus_Cyrl | Russian | Cyrl | ru | |
| srp_Cyrl | Serbian | Cyrl | | srp |
| swh_Latn | Swahili | Latn | sw | |
| tha_Thai | Thai | Thai | th | tha |
| tel_Telu | Telugu | Telu | te | |
| yor_Latn | Yoruba | Latn | yo | |
| zho_Hans | Chinese (Simplified) | Hans | zh | zho |

## A.4 ENTROPY-NORMALIZATION

Our decision to normalize our divergence metrics comes from the strong per-layer entropy patterns we see in the models (See Section 4.4 and Appendix A.5). KL-divergence, sometimes referred to as "relative entropy", is highly sensitive on entropy and therefore so is JS-divergence. With entropy decreasing across model layers near-monotonically, using simple JS-divergence meant our plots were overshadowed by the trends in entropy and therefore choose to control for it. This divergence metric properly compares divergence between scenarios when both are flattened distributions (high entropy) or peaked distributions (low entropy).

We normalize JS-divergence by approximating the theoretical maximum divergence given the entropy of the two distributions. The normalization factor $F$ is computed as $\log E - H_{avg}$, where $E$ is the vector size (number of experts) and $H_{avg}$ is the average entropy of the two distributions. This normalization accounts for the fact that distributions with different entropy have different upper bounds for JSD. Concretely, using the same notation as Section 4.3:

$$D_{\mathrm{JS}}(\boldsymbol{q}^{(1)}||\boldsymbol{q}^{(2)}) = \frac{1}{2}(D_{\mathrm{JS}}(\boldsymbol{q}^{(1)}||\overline{\boldsymbol{q}}) + D_{\mathrm{KL}}(\boldsymbol{q}^{(2)}||\overline{\boldsymbol{q}})) \quad \text{where } \overline{\boldsymbol{q}} = \frac{1}{2}(\boldsymbol{q}^{(1)} + \boldsymbol{q}^{(2)}) \quad (6)$$

$$F = \log E - \frac{1}{2}(H(\boldsymbol{q}^{(1)}) + H(\boldsymbol{q}^{(2)})) \quad (7)$$

$$D_{\mathrm{H\text{-}JS}}(\boldsymbol{q}^{(1)}||\boldsymbol{q}^{(2)}) := \frac{1}{F} D_{\mathrm{JS}}(\boldsymbol{q}^{(1)}||\boldsymbol{q}^{(2)}) \quad (8)$$

As a reminder, $E$ is the size of $\boldsymbol{q}^{(1)}, \boldsymbol{q}^{(2)}$ and corresponds to the number of experts.

## A.5 ENTROPY PLOTS FOR ALL MODELS

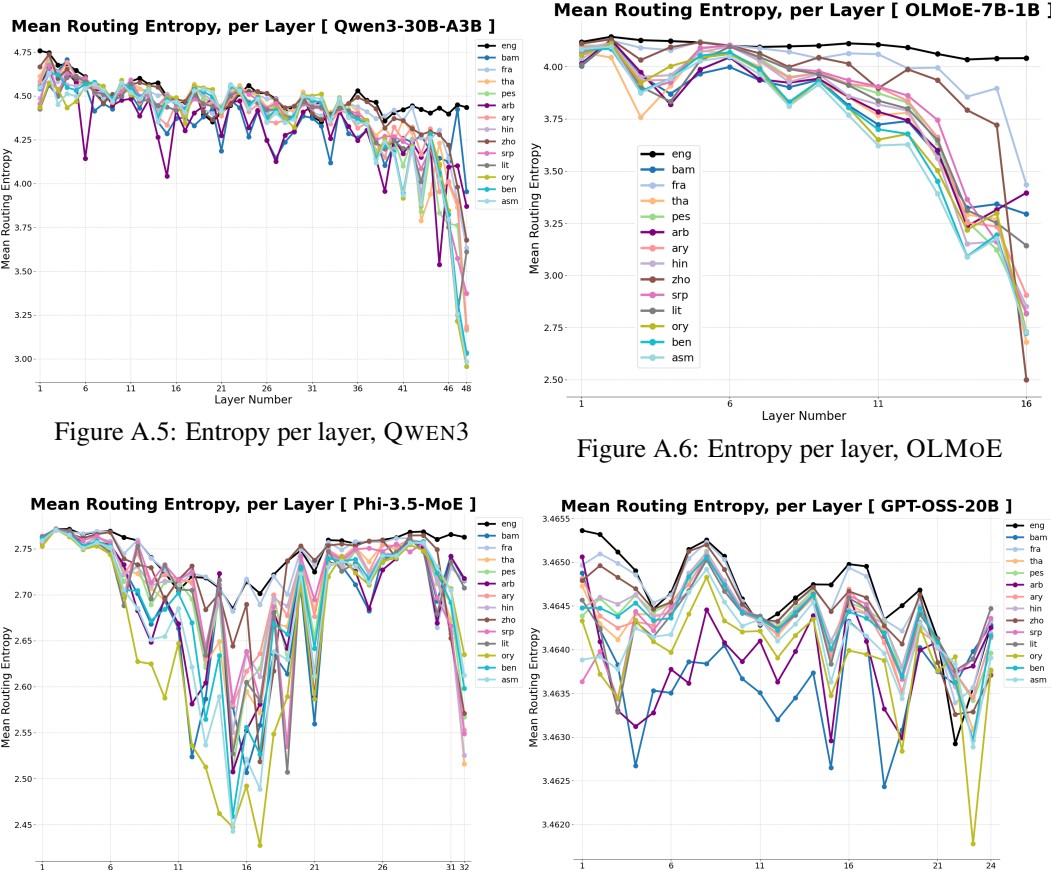

Figure A.5: Entropy per layer, QWEN3

Figure A.6: Entropy per layer, OLMOE

Figure A.7: Entropy per layer, PHI-3.5-MOE

Figure A.8: Entropy per layer, GPT-OSS

## A.6 MINUTE DETAILS FOR HARD INTERVENTION

Similar to Fayyaz et al. (2026), we guard against the potential of breaking the top-k logic by adding a random perturbation in Equation 5. In the case where the the number of experts selected for force-activation is not less than the model's experts-per-token (k), the LLM would otherwise throw an error trying to pick k experts from $> k + 1$ experts with exactly the same maximum value (at least with our vLLM implementation). Tiny random perturbation ensures the values are not identical and the top-k can be chosen. We note, however, the activation of the experts are not technically guaranteed under this hard-intervention. However, this ends up being inconsequential as, empirically, we find that hard-intervening on so many experts at once derails the model anyways. This even holds true for PHI-3.5-MOE where only 2 experts are activated.

A.7 EXAMPLE PLOT FOR DIFFERENCE IN RELATIVE FREQUENCY OF ACTIVATION

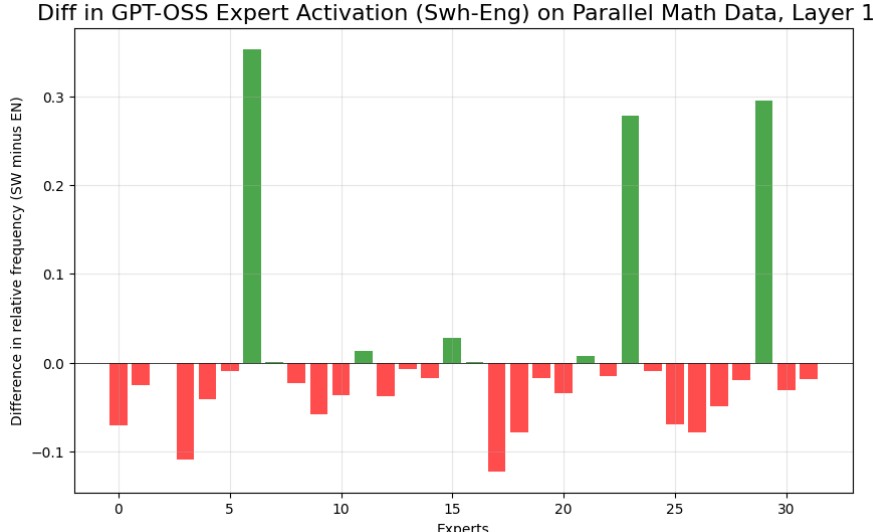

Figure A.9: Example of visualization of our metric for selecting specialized experts. This is the $\Delta$ described in Section 5.2, the difference in activation relative frequency as defined in Equation 3. Here we display an example; $\Delta$ between Swahili and English (on FLORES).

Positive values mean the expert is activated more in Swahili than English. Since the distribution has to be mean-zero, we see most experts have slightly negative value while a small few (in this case 3) are strongly positive. This type of graph was almost always the case, across models, model layers, languages and domains when comparing to the FLORES English set. This allowed for clear selection of language- or domain-specialized experts using the threshold $\tau$.

A.8 LOCATION OF SPECIALIZED EXPERTS

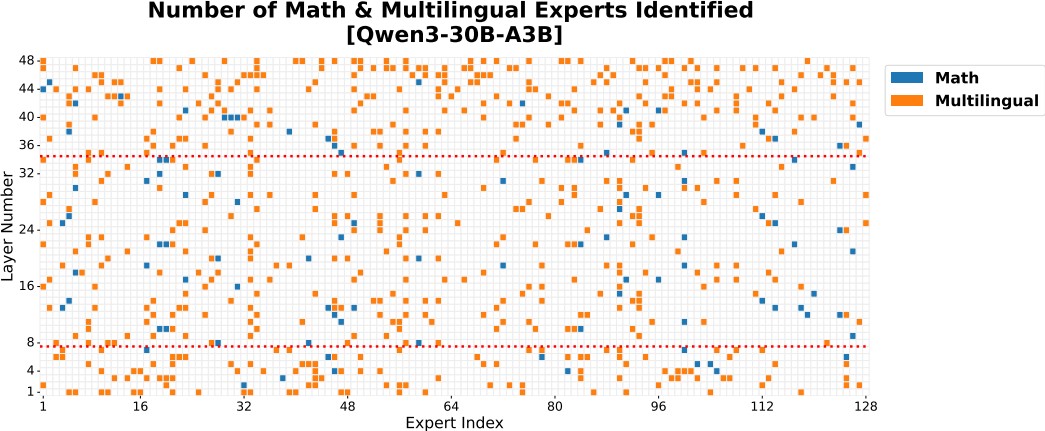

Figure A.10: A different view from Figure 5 of the location of identified experts, displaying the lack of overlap. Here for QWEN3, $\tau = 0.3$ used for identification. Layer numbers are from bottom to top and expert numbers are an ID, it is numerically meaningless; simply meant to display the sparsity. The red horizontal bars delimit the region in which we intervene.

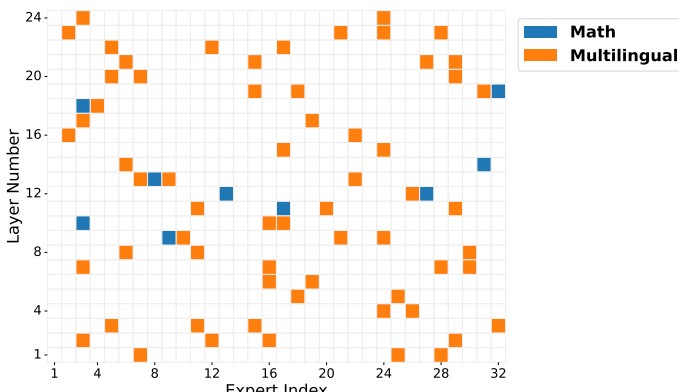

Figure A.11: Now for GPT-OSS-20B, also with $\tau = 0.3$ used for identification. Compared to QWEN3, GPT-OSS had less layers and less experts per layer but the sparsity is similar. Regardless, there is an intersection of 0 between the 9 math-specialized experts and 63 multilingual-specialized experts.

