# OpenReview forum: "Multilingual Routing in Mixture-of-Experts"
_ICLR.cc/2026/Conference — ICLR 2026 Poster_

### Official Review · Reviewer_pESw · 2025-10-17

**Soundness:** 3
**Presentation:** 3
**Contribution:** 3
**Rating:** 6
**Confidence:** 3

**Summary:**

This paper studies the behavior of different languages ​​in the routing mechanism of a mixture-of-experts (MoE) model in multilingual scenarios and proposes a method to improve multilingual performance by intervening in routing. This paper proposes a soft intervention approach, which adjusts expert scores before routing, and a hard intervention approach, which forcibly activates certain experts. Experimental results demonstrate that the proposed method effectively improves the model's performance in other languages.

**Strengths:**

1. The authors conduct an in-depth exploration of the behavior of different languages ​​in the MoE's large language model and summarize four findings.

2. There is a close connection between the proposed method and the findings.

3. The proposed method can improve the performance of the model in other languages ​​without further training.

**Weaknesses:**

1. Lack of further explanation for how the proposed method work.

2. Experiments lacking qualitative analysis.

**Questions:**

1. The authors do not explain why closer approximation to English leads to improved performance. Can similar effects be observed by making a low-resource language closer to a high-resource language?

2. There are no qualitative experiments demonstrating the inference process after intervention, or whether using an English expert leads to changes in the output language during inference.

3. The authors' proposed method involves many hyperparameters. To what extent do the choices of these hyperparameters affect the results?

---

> ### Author Response · Authors · 2025-11-20
> **Response to pESw**
>
> Thank you for review; from it we have gotten actionable feedback to improve our paper. We hope to address some of the points you make:
>
>
> Weaknesses:
> 1. We appreciate this feedback and apologize if the methodology was not sufficiently clear. Assuming that by the “proposed method” you mean the intervention, we believe the simplicity of the intervention method is one of its strengths. As described in Section 5.3, we intercept the logits outputted by the router and modify them (as described in Formula 4&5) before the model then selects and calls experts. As alluded to in the Reproducibility, this is implemented in vLLM, by customizing the forward() pass of the MoE module. Otherwise, we suspect the reviewer may be looking for conceptual explanation for this logit manipulation, which is that by increasing targetted logits as in Formulas 4&5 we are encouraging (or forcing) the logit associated with a particular expert to be in the top-k experts that are subsequently activated. We will enhance the text in Section 5.3 to explicitly articulate this conceptual overview, ensuring the clarity matches the technical detail.
> 1. It is true that we did not provide qualitative results but we note that the scores overall on both evaluation benchmarks did not fundamentally change. A model change that improves performance on a test set from 68.5% to 70.2% likely does not drastically corrupt the model’s generation capabilities. We note, however, that any decrease in generation capabilities would likely affect the MGSM scores negatively (as could be happening with some of our baseline interventions). GSM is generative task that requires model reasoning and a natural language answer. We have looked at a few samples as a sanity check on numerous occasions (they looked fine), but believe these small changes in benchmark performance do not warrant a full qualitative assessment. That being said, we can provide some sample model responses: such as a MGSM sample in the appendix where the model originally answered the question wrong before and after intervention, answered the question right.
>
> Questions:
> 1. We choose English because this is the language that constitutes the highest % of training data and also the language in which the model performs best on these target benchmarks to begin with. Previous works find that these LLMs "think" using language-universal representations, but that this representation space is still quite English-centric [1][2]. However, we note that we did look at routing divergence with respect to other high resource languages (e.g. Chinese, French) and find very similar patterns. We omitted this detail last minute to fit into 9 pages but since you find it relevant, will add it back. We did not, however, experiment with other languages for intervention.
> 1. (Addressed above)
> 1. In retrospect, we acknowledge our discussion in the Sensitivity to Target Layers subsection lacks clarity on this subject. What we meant to highlight here is that in a reasonable range of these hyperparameters we introduce (selection threshold, hard/soft, lambda), the differences are not all too different. However, once you change the location (layers) of intervention, performance degradation happens very quickly (which you would expect for random interventions of this magnitude).
>
> Concrete steps we will take from your feedback, enabled by additional page:
> 1. Enhance the text in Section 5.3 to better explain how we are forcing (or encouraging) certain experts to be activated, ensuring the clarity matches the technical detail.
> 1. Add a sample generation for an MGSM question the intervention led the LLM to the correct answer, to the appendix.
> 1. Add back a sentence detailing that similar routing divergence graphs/findings were gotten when centering Chinese, French, Spanish, instead of English.
> 1. Expand and clarify the  “Sensitivity to Target Layers” section about hyperparameters to emphasize significance of connection between intervention results and earlier findings
>
> We thank you in advance for reading our response and look forward to improving our paper.
>
> [1] Wendler et al., 2023
>
> [2] Wu et al., 2025

---

> ### Comment · Reviewer_pESw · 2025-11-20
>
> Thank you for your reply.
>
> Q1 refers to the theoretical analysis behind this phenomenon. For example, different languages ​​have different grammar (e.g., subject-verb-object order), so why would bringing them closer to a language with richer resources lead to improvement?
>
> And, perhaps the authors should check their references?

---

> ### Author Response · Authors · 2025-11-22
> **Response to pESw's comment**
>
> Q1: Thanks, we understand the question better. Hopefully this response can help explain our intuition:
> - Why our method works: Our intuition built off previous work and our routing analysis is that the first few transformer layers and last few handle language-specific tasks, while the middle layers operate on abstract concepts. Previous work finds low-level linguistic differences, such as (e.g., subject-verb-object order), are primarily processed in the model's early and final layers [1]. Mechanistic interpretability works describe the process in the first few layers are responsible for "detokenization" [2], where raw tokens are combined into abstract concepts (e.g. building a representation for “social security” instead of “social”, “sec_”, or “_urity” individually) [3]. Previous work finds that the resulting concept space of the middle layers is reasonably language-universal (i.e. all languages are mapped to it), even though it is still English-centric, which can be seen most easily with logit lens [4][5]. In these middle layers, English likely accesses the best-trained task-specific experts (for math or medicine). For low-resource languages, the model may not have effectively learned to align its tokens with these shared universal concepts, causing it to miss out on the best task experts. That’s why by encouraging/forcing the model to activate those (potentially) best experts, the benefits outweigh the fact we are interfering with the models extensive-trained routing process.
> - Why baselines don’t work: Under this intuition, intervening in the early or final layers disrupts the language-specific mechanisms (like part-of-speech or grammar handling as you note) that the model needs to properly process the input. When we intervene in these layers, performance rapidly degrades.
>
> We additionally note that it's not fully homogenized routing. Even our "hard intervention" only forces 1 or 2 experts into the top-K selection (where K is typically 4 or 8). The router still produces a full logit distribution for every token, and it chooses most experts as before. The intervention is a targeted nudge to ensure the task-relevant information is considered.
>
>
> We hope this helps articulate our intuition and if you believe so, we will work it into our paper as ensuring the reader understands these conceptual takeaways is obviously very important for our work. If not, we would be happy to keep discussing to improve the presentation of our findings.
>
> [1] https://arxiv.org/pdf/2401.04088 Jiang et al., 2024
>
> [2] https://arxiv.org/abs/2305.01610 Gurnee et al., 2023
>
> [3] https://www.alignmentforum.org/posts/iGuwZTHWb6DFY3sKB/fact-finding-attempting-to-reverse-engineer-factual-recall Nanda et al., 2023
>
> [3] https://aclanthology.org/2024.acl-long.820.pdf Wendler et al., 2023
>
> [4] https://openreview.net/pdf?id=FrFQpAgnGE Wu et al., 2025

---

### Official Review · Reviewer_JWgw · 2025-10-28

**Soundness:** 4
**Presentation:** 4
**Contribution:** 4
**Rating:** 8
**Confidence:** 5

**Summary:**

This work conducts an in-depth study on the routing pattern  of MoE models on multi-lingual data. Authors conclude five findings via visualization and one more finding about how to leverage the observations to further improve the model.

**Strengths:**

1) The writing is very clear. It is great to highlight the findings in this paper as it is indeed a bit easy to get lost when reading the detailed analysis and the description of experimental setups.
2) The finding 6 is a good evidence helping support the correctness of the findings via visualisation. It is also interesting that simply changing the routing decision by aligning with english can improve the performance.
3) Consider MoE is becoming a default choice of most LLMs. It indeed motivates the community to rethink the strength and weakness of using such a sparse model. For the long term purpose, as the compute is getting cheaper and cheaper, we might roll back to dense for better and more robust universal representations.

**Weaknesses:**

1) It would be helpful to provide more insights about how these findings are connecting to each other, and maybe draw a figure to explain this in the final draft.
2) Minor: To my best knowledge, this work (https://arxiv.org/abs/2402.01739) should be one of the earliest work taking a closer look at this problem. I understand that the model has been a bit too weak and some designs have been outdated, but it would be helpful to discuss this work as a reference properly.

**Questions:**

NA

---

> ### Author Response · Authors · 2025-11-20
> **Response to JWgw**
>
> Thank you for the encouraging assessment of our paper and its strengths. We address your weaknesses:
>
> Weaknesses:
> 1. This is valuable feedback, thank you. All the findings are highly interconnected and caused by multilingual (& task) specializations of experts, and we can add sentences for each explaining this intuition further (thanks to the additional page). Can highlight quickly how we believe they’re interconnected and what we can emphasize more.
> - Routing entropy likely drops so much in the last layers for multilingual text because of the language-specialization in the last few layers and how confident the router is to send the tokens to . For English, the model has many experts to choose from (e.g. maybe 100 out of 128), while for Thai, there’s maybe 1-2 experts that know how to generate Thai tokens ⇒ higher routing confidence.
> - The above is interconnected with intra-sequence routing. If there are only 1-2 experts for generating Thai, it will route as such for most/all tokens in a Thai sequence, while in English maybe there’s more options. It is surely not as clean as this, but this is our intuition.
> - In general, the other findings are interconnected by the principle that the model’s ability to do math/hold knowledge is stored in a language-agnostic manner, and the ability to use the language-specialized experts in the early/late layers to map to/from representations that allow it to access such language-agnostic parameters.. In the middle layers, if an expert is language-specialized, it’s likely it’s just an expert that is not as useful for solving the question, while the useful stuff is stored in a language-agnostic way.
> 2. Thank you, Section 4.1 of this paper, notably, will help improve our introduction or MoE preliminaries.
>
> Concrete steps we will take to improve our paper, enabled by the additional page:
> 1. Discuss the OpenMoE paper in our MoE preliminaries
> 1. Better explain the inter-connectedness of the findings in Section 4.
>
> Thanks and we look forward to improving our paper.

---

> > ### Comment · Reviewer_JWgw · 2025-11-27
> >
> > Thanks. Sounds good to me. good luck

---

### Official Review · Reviewer_gR2T · 2025-10-31

**Soundness:** 2
**Presentation:** 3
**Contribution:** 2
**Rating:** 2
**Confidence:** 4

**Summary:**

Paper presents a descriptive analysis of how MoEs process multilingual data and investigates the router patterns of tokens for different languages and presents the trends like tokens are routed in language-specific ways in the early and late layers, but exhibit significant cross-lingual alignment in the middle layers, mirroring phenomena observed in dense models. The paper's core contribution is demonstrating a strong correlation between a model's performance in a given language and the similarity of its token routing to English in these middle layers. Paper then presents a simple intervention to leverage this phenomena to sterr model to activate English-identified task-specific experts for non-English inputs, however this method yields only a marginal boost in performance.

**Strengths:**

1. The paper is clearly written and easy to follow, with effective visualizations and well-designed, information-theory informed probes to analyze routing patterns in multilingual data. The analysis  using an entropy-normalized JS divergence metric to study routing divergence, complemented by routing entropy and consistency analyses is good to me.

2.The work includes comparisons across model scales and multiple languages, enhancing the generality of the findings.

3. . The figures, especially Figure 1 and Figure 2, are excellent visualizations that clearly communicate the central hypothesis of the U-shaped routing divergence and its correlation with model performance.

4. The finding of complete modularity between task-specialized and language-specialized experts (Finding 5) is particularly strong and clean.

**Weaknesses:**

1. The proposed method improves performance by only 1–2%, which is not practically meaningful. The analysis and intervention seem too complex for such minimal payoff, which makes it hard to justify the paper’s overall significance.

2. The proposed “steering” approach feels like a post-hoc, brute-force fix. Forcing a model to use certain experts is more of a hack than a genuine solution. The method shows that steering can slightly help, but not that it’s a meaningful or scalable way forward.

3. The observation of middle layers of LLMs form a shared language space has already been well established for dense models. Showing that MoEs follow the same pattern is interesting but not surprising. The paper essentially confirms existing ideas rather than offering something genuinely new.

**Questions:**

1. The paper suggests a causal link between routing alignment and multilingual performance, but the evidence feels somewhat weak. A stronger case would involve showing that performance systematically improves as the degree of steering increases, or that steering towards unrelated experts consistently hurts performance. Is there enough evidence to make a strong causal claim here?

2. In Table 2, the intervention appears to slightly degrade performance in English. How do you interpret this? Could it be that forcing a fixed subset of experts interferes with the model’s naturally optimized routing for its primary language?

3. Regarding Finding 5, the clean separation between multilingual and task-specific experts is quite striking. Does this separation remain consistent across different expert selection thresholds (τ)? If the threshold were lowered, could there be experts that participate in both language and task representations, and what might that mean for the design or effectiveness of your intervention?

---

> ### Author Response · Authors · 2025-11-17
> **Addressing Weaknesses from gR2T**
>
> Thank you for your in-depth review, we are grateful for the care you took. We appreciate the positive comments on our analysis, visualizations, and the strength of Finding 5. However, we hope we can also address many of your concerns below.
>
> Weaknesses:
>
> 1 & 2. We want to push back on the framing of our intervention results as not practical/scalable:
> - Purpose of the Experiment: The intervention's goal was not to deliver a new SOTA method to productionize, but to empirically validate our analysis and hypothesis. By successfully steering the model, we show that our findings are not unimportant artifacts, but rather demonstrate a verifiable model mechanism for dealing with multilingual text. In this sense, we argue it’s much more than a hack because of how interlinked it is with our deep analysis of routing patterns & resulting intuition.
> - That being said, a 1-2% gain is substantial, considering these are simple, test-time interventions applied to SOTA, extensively-trained models. The fact that by messing with the router we are able to cause positive impact is notable in this context.
> - The consistency of this improvement across three different models, two tasks, and 15+ languages is more significant than its absolute magnitude, demonstrating a reliable underlying mechanism
>
> So while this cheap method’s impact is minimal in magnitude, it suggests there is tremendous potential for improving model cross-lingual alignment during training or otherwise.
>
> 3. We agree—our intuition was that a shared representation space would also exist in MoEs—however this still needed to be explored given the completely different computational paradigm of MoEs. Our contribution is in showing how this is mechanistically implemented and modularized in sparse MoE architectures. And using the mechanism of expert-routing, we find this pattern in a much stronger, numerical way than has been done before (e.g. the high interpretability of Fig.1 and Fig. 2). Previous method use prompting [1], logit lens [2], dense neuron activations [3], or SFT gradients [4] are much less direct and/or noisy. Even so, we argue our analysis extends far past this simple finding: we {a} reveal a very clean, structural separation between task-specialized and multilingual experts (Fig. 5 & Finding 5), {b} perform interventions to validate that it’s more than just a correlation, and {c} offer numerous other related findings that are a consequence of multilingual specialization. Based on your feedback we will detail deeper in the Related Works section how our paper contributes to this previous research.
>
> [1] prompting methods: Shi et al., 2023; Zhang et al., 2024; Yong
> et al., 2025
>
> [2] logit lens analyses: Wendler et al., 2023; Wu et al., 2025
>
> [3] dense neuron activations analyses: Kojima et al., 2024; Tang et al., 2024; Alabi et al., 2024
>
> [4] gradient-based analysis: Bandarkar et al., 2025

---

> > ### Author Response · Authors · 2025-11-17
> > **Answering Questions and proposing improvements, from gR2T**
> >
> > Questions
> >
> > 1. We did a large number of baselines: changing the layers included in the intervention, fully random interventions, or simply deactivating multilingual experts without moreso steering towards the English experts. All this led to rapid performance degradation, as discussed L384-391. We reason that further increasing the magnitude of steering does not improve because it destabilizes the routing & aggregation process too much. In hard-intervention, we are already activating experts activated on 30-80% of tokens in English on all tokens. In soft-intervention, large lambdas risk sharpening the weight distributions across the experts, disrupting the model's typical process for aggregating numerous experts. The model would likely have to be retrained for an even stronger intervention to be possible.
> > 2. There is some degradation in English, notably for Phi on MGSM, but we think this is not super notable. The reason is that, in English, we are encouraging the router to route more strongly towards the experts it was already activating. There’s no reason this should improve the model, so a slight decrease seems reasonable.
> > 3. Your intuition is correct, if you lower the threshold and select a larger amount of experts you do see some overlap, but it remains surprisingly low (if 𝜏=0.2, there is still <10 experts identified as both task- and multilingual-specialized. With this sort of identification, lowering the threshold increases your odds of “false negatives” (experts that are not actually specialized for this, but selected because of variance). We believe that’s why for all models 𝜏$\geq$0.25 works for intervention. This was one of the last details we cut to fit page limit, we can now add this detail back.
> >
> >
> > Concrete steps we will take from your feedback, enabled by additional page:
> > - Expand related works to better explain how this paper further contributes on top of past multilingual work
> > - In intro & conclusion, ensure the method is not interpreted as proposing a viable solution but rather one that validates the findings and “motivate future work on methods that enhance cross-lingual routing alignment and the sharing of specialized experts” (L483).
> > - Expand discussion of results to ensure the reader understands the many baselines we ran & things we tried that didn’t work, to reinforce the conclusions.
> >
> >
> > We thank you in advance for reading our long response and we hope that you reconsider the initial review given.

---

> > ### Comment · Reviewer_gR2T · 2025-11-24
> > **Thank you**
> >
> > I appreciate the authors’ careful and thoughtful responses. Although the performance gains are modest, I value the use of MoE in the multilingual setting, and I am raising my score to 6. I would also appreciate further elaboration on how these minimal gains might be improved across more languages in future work.

---

> > > ### Author Response · Authors · 2025-11-25
> > > **Requested further elaboration by reviewer gR2T**
> > >
> > > Thank you for reviewing our response.
> > >
> > > While we leave it to future work, ideas for how to improve routing across more languages:
> > > - incorporating more parallel translation data for training, which has shown to align representations
> > > - auxiliary losses that incentivize more similar routing across languages (perhaps using parallel data in a contrastive setup)
> > >  - multilingual training with frozen middle layers to force mapping from multilingual in/outputs to/from semantic hub

---

### Official Review · Reviewer_rfTm · 2025-11-01

**Soundness:** 2
**Presentation:** 3
**Contribution:** 3
**Rating:** 6
**Confidence:** 4

**Summary:**

The paper investigates multilingual behavior in MoE LLMs. Using models such as Qwen3-30B, Phi-3.5-MOE, GPT-OSS-20B, and OlmoE, the authors analyze expert routing patterns across languages and layers. They find that early and late layers are language-specific while middle layers exhibit cross-lingual alignment. They propose inference-time “routing interventions” that encourage the activation of experts often used for English, reporting 1–2% improvements on multilingual benchmarks (MGSM and Global-MMLU).

**Strengths:**

- Timely topic: Multilingualism in MoE architectures is underexplored; the paper touches on a relevant question for LLM scaling.
- Comprehensive dataset use: Employs multiple evaluation datasets (FLORES, BELEBELE, MGSM, Global-MMLU).
- Interpretability focus: The routing-divergence visualizations and entropy analyses are informative.
- Reproducibility: The methodology is relatively transparent, with sufficient detail to replicate the experiments.

**Weaknesses:**

- Marginal contribution: The reported 1–2% improvements are minor and lack depth of analysis. No theoretical insight or convincing causal mechanism is demonstrated. The “steering” interventions are heuristic and depend heavily on ad hoc hyperparameter tuning (λ, τ, and layer ranges). No systematic ablation or generalization evidence is provided.

- Interpretation overreach: The claim of “causal relationships” between routing alignment and multilingual performance is not empirically supporte (correlation is misinterpreted as causation).

- Lack of broader impact or insight: The paper neither improves multilingual modeling methods nor contributes significant interpretability tools.

**Questions:**

- The findings that early and late layers are language-specific, while middle layers exhibit cross-lingual alignment, also hold for dense-model architectures, as noted by the authors. You may want to reference this recent paper on the topic: https://aclanthology.org/2025.findings-acl.1385
- Overall, my assessment is toward acceptance, as it is a timely topic.

---

> ### Author Response · Authors · 2025-11-18
> **Response to rfTm**
>
> Thank you for review; from it we have gotten actionable feedback to improve our paper. We hope to address some of the weaknesses you list:
> 1. We want to push back on the framing of our intervention results as not minor/lacking depth. While this method’s impact is minimal in magnitude, it suggests there is tremendous potential for improving model cross-lingual alignment during training or otherwise.
> - Purpose of the Experiment: The intervention's goal was not to deliver a new SOTA method to productionize, but to empirically validate our analysis and hypothesis. By successfully steering the model, we show that our findings are not unimportant artifacts, but rather demonstrate a verifiable model mechanism for dealing with multilingual text. In this sense, we argue it has tremendous theoretical insight:
> - In this context, a 1-2% gain is substantial, considering these are simple, test-time interventions applied to SOTA, extensively-trained models. The fact that by messing with the router we are able to cause positive impact is very notable. The consistency of this improvement across three different models, two tasks, and 15+ languages is more significant than its absolute magnitude, demonstrating a reliable underlying mechanism.
> - We did a large number of baselines: changing the layers included in the intervention, fully random interventions, or simply deactivating multilingual experts without moreso steering towards the English experts. All this led to rapid performance degradation, as discussed L384-391, but we will make this much more clear to further emphasize the cleanliness of the analysis findings → intervention results connection.
> - With respect to hyperparameters, in retrospect, we acknowledge our discussion in the Sensitivity to Target Layers subsection lacks clarity on this subject. What we meant to highlight here is that in a reasonable range of these hyperparameters we introduce (selection threshold, hard/soft, lambda), the differences are not all to different. However, once you change the location (layers) of intervention, performance degradation happens very quickly (which you would expect for random interventions of this magnitude). But yes, it requires post hoc tuning because each router is highly sensitive (and has different configs) so finding this range of hyperparameters requires trial and error.
> 2. We were extremely careful about claiming causality — we only use the word “causal{...}” once at L482. That being said, our inference-time interventions provide controlled experiments that show that encouraging routing more similarly to English leads to positive outcomes. As a result, we argue that it is appropriate and not an overstatement in the conclusions & future work to state that this “suggests a causal relationship” (L482) and argue that this should prompt future work to test enhancing alignment (probably would require a training method). We would be happy to change the wording of this sentence to appease the reviewer’s concerns, but believe we did not overreach.
> 3. We argue we have significantly contributed to understanding model mechanisms for multilingual processing. Our method of using routing data on parallel data is novel and intuitive, and uncovers highly interpretable patterns. Previous method in dense LLMs that use prompting [1], logit lens [2], dense neuron activations [3], SFT gradients [4], and embeddings similarity [5] are much less direct and/or noisy. While we don’t provide a “tool” (since the method requires forward passes), we provide all the necessary means to reproduce and apply it to other models/data. Furthermore, we find numerous important findings and produce very clear visualizations with these interpretability mechanisms.
>
>
> Concrete steps we will take from your feedback, enabled by additional page:
> 1. Expand and clarify (1) discussion of baselines & intervention alternatives (L384-391) and (2) “Sensitivity to Target Layers” section about hyperparameters to emphasize significance of connection between intervention results and earlier findings
> 1. Expand related works to better explain how this paper further contributes on top of past multilingual work
> 1. In intro & conclusion, ensure the method is not interpreted as proposing a viable solution but rather one that validates the findings and “motivate future work on methods that enhance cross-lingual routing alignment and the sharing of specialized experts” (L483).
> 1. The MEXA paper you link is indeed highly relevant and analogous work that we will incorporate in numerous places.
>
>
> We thank you in advance for reading our (long) response and look forward to improving our paper .
>
>
>
> [1] prompting methods: Shi et al., 2023; Zhang et al., 2024; Yong
> et al., 2025
>
> [2] logit lens analyses: Wendler et al., 2023; Wu et al., 2025
>
> [3] dense neuron activations analyses: Kojima et al., 2024; Tang et al., 2024; Alabi et al., 2024
>
> [4] gradient-based analysis: Bandarkar et al., 2025
>
> [5] embedding similarity: Kargaran et al., 2025

---

> > ### Comment · Reviewer_rfTm · 2025-11-19
> >
> > Thank you for your response. I will raise my score to 8. Hopefully, more multilingual NLP research will appear at ICLR.

---

### Author Response · Authors · 2025-12-04
**Summary of Discussion Phase**

For the benefit of the AC, we try and summarize the initial reviews and the resulting discussion phase below. The end result is that **all four reviewers concluded that the paper’s soundness, presentation, and contribution were largely sufficient for conference acceptance**. We summarize some key points:
### Soundness
- Reviewers highlighted our interpretability methodology, its explanation, and the resulting findings as a central strength of the paper (pESW, gR2T, rfTM).
- Reviewers (rfTm, gR2T) questioned the "causal" claims, but we argued our writing was extremely careful with this wording, and while our work suggests causal relationship, we stopped short of claiming we had proved as much. We leave more methodical (i.e. training-based) experiments for this concrete relationship to future work.
### Presentation
- Figures 1 and 2 (mentioned by gR2T) and the overall clarity of the writing (mentioned by rfTm, gR2T, JWgw) are highlighted as strengths.
- Reviewer pESw requested a clearer theoretical explanation for why aligning routing to English leads to performance gains in other languages, which we addressed with a detailed conceptual response.
- Our biggest takeaway from this discussion is the need to better organize how we presented the baseline and hyperparameter-tuning experiments. It appears the results from these experiments did not effectively convey the degree to which they validated our conclusions. This relates to comments from rfTm, gR2T, and pESW.
### Contribution
- The novelty and depth of the interpretability analysis are highlighted by all reviewers. In addition, reviewers highlight the timeliness of such an analysis for MoEs (rfTm, JWgw).
- However, some (rfTm, gR2T) questioned the contribution on top of previous works that already find LLM’s “think in English”. We argued that beyond providing MoE-specific findings (e.g. routing entropy), our method is less noisy and/or more direct than previous approaches and the patterns we uncover are more interpretable & convincing than ever.
- Perhaps the most important shortcoming initially highlighted by reviewers (rfTm, gR2T) was that the small improvements from our intervention meant this method was insignificant and or impractical. We pushed back by saying this seems to miss the point: the intervention's goal was not some method to be implemented but empirical validation of our interpretability findings (a "verifiable model mechanism"). We emphasized the consistency across models/tasks/languages but also that in the context of how we intervened, the magnitude also is notable.


# Rebuttal Revisions
We have submitted a rebuttal revision, to be cleaned later for camera-ready in case of acceptance. The modifications (in red in the paper) we have made are:
- Expand and clarify (1) discussion of baselines & intervention alternatives (Section 5.4 and 6) and (2) “Sensitivity to Target Layers” section about hyperparameters to emphasize significance of connection between intervention results and earlier findings
- Expand discussion of results to ensure the reader understands the many baselines we ran & things we tried that didn’t work, to reinforce the conclusions​​. In addition, improve discussion of intuition
- Expand related works to better explain how this paper further contributes on top of past multilingual work. Plus, add references to MEXA and OpenMoE papers.
- In intro & conclusion, ensure the method is not interpreted as proposing a viable solution but rather one that validates the findings and “motivate future work on methods that enhance cross-lingual routing alignment and the sharing of specialized experts” (L483, now L523).
- Add a sample generation for an MGSM question the intervention led the LLM to the correct answer.
- Add back a sentence detailing that similar routing divergence graphs/findings were gotten when centering Chinese, French, Spanish, instead of English.
- Better explain the inter-connectedness of the findings in Section 4.

---

### Meta-Review · Area_Chair_XY4J · 2026-01-06

**Summary:**

The initial reviews were mixed (range from a "reject" from gR2T to an "accept" from jWgw), however the discussion phase was very productive. The final decision to accept is driven by the paper's strong interpretability analysis of MoE routing for multilingual modeling and the empirical validation that steering routers toward "language-universal" middle-layer experts to improve multilingual performance, though the gains are modest.

**Reviewer Concerns:**

The reviewer concerns were properly address, specifically:

gR2T:
* the proposed method was a "post-hoc, brute-force fix" and argued a 1-2% gain was too marginal: the author clarified the method is an inference-time intervention, it's light wight and mainly for illustrating the observed interpretations rather than to build SoTA.
* questioned the causality of the observed correlations and the intervention slightly degraded English: the authors carefully wording it as "suggests a causal relationship" for future verification.

rftM:
* marginal contribution (1-2% gains): similar as above, the gains are to empirically validate the analysis.
* causal mechanism, overclaimed causal relationship: the authors provided more baselines and softened the wording

pESw:
* lack of theoretical justification why it works: the author provided a detailed explanation of "language-universal" representation in middle layers.
* qualitative analysis: the authors plan to add samples to the paper

JWgw:
* minor concerns on better visualization and missing citation: the authors promised to add the missing reference and improve visualization.

**Reviewer Scores:**

gR2T: initial 2, updated to 6 after rebuttal, no further change
rftM: initial 6, updated to 8 after rebuttal, no further change
JWgw: maintained 8, no further change
pESw: maintained 6, no further change

---

### Decision · Program_Chairs · 2026-01-26

Accept (Poster)